# Neutralizing antibodies from prior exposure to dengue virus negatively correlate with viremia on re-infection

Anbalagan Anantharaj[1], Tanvi Agrawal [1], Pooja Kumari Shashi[1], Alok Tripathi[1,7], Parveen Kumar[1,7], Imran Khan[1], Madhu Pareek[1], Balwant Singh[1], Chitra Pattabiraman[2], Saurabh Kumar[1], Rajesh Pandey[3], Anmol Chandele[4], Rakesh Lodha[5], Stephen S. Whitehead [6] & Guruprasad R. Medigeshi [1✉]

## Abstract

**Background** India is hyperendemic to dengue and over 50% of adults are seropositive. There is limited information on the association between neutralizing antibody profiles from prior exposure and viral RNA levels during subsequent infection.

**Methods** Samples collected from patients with febrile illness was used to assess seropositivity by indirect ELISA. Dengue virus (DENV) RNA copy numbers were estimated by quantitative RT-PCR and serotype of the infecting DENV was determined by nested PCR. Focus reduction neutralizing antibody titer (FRNT) assay was established using Indian isolates to measure the levels of neutralizing antibodies and also to assess the cross-reactivity to related flaviviruses namely Zika virus (ZIKV), Japanese encephalitis virus (JEV) and West Nile virus (WNV).

**Results** In this cross-sectional study, we show that dengue seropositivity increased from 52% in the 0–15 years group to 89% in >45 years group. Antibody levels negatively correlate with dengue RNAemia on the day of sample collection and higher RNAemia is observed in primary dengue as compared to secondary dengue. The geometric mean $FRNT_{50}$ titers for DENV-2 is significantly higher as compared to the other three DENV serotypes. We observe cross-reactivity with ZIKV and significantly lower or no neutralizing antibodies against JEV and WNV. The $FRNT_{50}$ values for international isolates of DENV-1, DENV-3 and DENV-4 is significantly lower as compared to Indian isolates.

**Conclusions** Majority of the adult population in India have neutralizing antibodies to all the four DENV serotypes which correlates with reduced RNAemia during subsequent infection suggesting that antibodies can be considered as a good correlate of protection.

## Plain Language Summary

India is one of the hotspots of dengue infection. The objective of the study was to assess whether having previous exposure to dengue virus could influence how the body will respond to repeat infections with dengue virus. Here, we analysed samples from febrile patients to measure the amount of dengue virus genetic material in the blood, the type of virus and the amount of antibodies, which are proteins produced by the host in response to dengue virus infection. The majority of patient samples demonstrated the capability to restrict all four types of dengue virus in circulation within the country, but reduced capacity to restrict when it comes to international dengue virus types. These data will help to inform future dengue vaccine design and clinical studies in India.

[1] Bioassay laboratory and Clinical and Cellular Virology lab, Translational Health Science and Technology Institute, Faridabad, Haryana, India. [2] Independent researcher, Bengaluru, India. [3] INtegrative GENomics of HOst-PathogEn (INGEN-HOPE) laboratory, Division of Immunology and Infectious Disease Biology, CSIR-Institute of Genomics and Integrative Biology, Delhi, India. [4] ICGEB-Emory Vaccine Center, International Center for Genetic Engineering and Biotechnology, Aruna Asaf Ali Marg, New Delhi, India. [5] Department of Pediatrics, All India Institute of Medical Sciences, Ansari Nagar, New Delhi, India. [6] Laboratory of Viral Diseases, National Institute of Allergy and Infectious Diseases, National Institutes of Health, Bethesda, MD 20892, USA. [7] These authors contributed equally: Alok Tripathi, Parveen Kumar. ✉email: gmedigeshi@thsti.res.in

Dengue virus (DENV) is a single-stranded positive-sense RNA virus belonging to the genus flavivirus in the family *Flaviviridae*. It causes dengue, a mosquito-borne febrile infection which is primarily transmitted to humans by *Aedes aegypti* and *Aedes albopictus* mosquitoes[1,2]. Dengue is of immense public health importance with an estimated 3.9 billion global population at risk, annual 96 million clinical cases and 20,000 deaths[3]. The disease is endemic in more than 125 countries with most cases being reported from the South-East Asia, Americas, and Western Pacific regions of WHO[4, 5]. In India, dengue is endemic in almost all states and is the leading cause of hospitalization during annual outbreaks[6, 7]. There are four serologically and genetically distinct serotypes, DENV-1 to DENV-4, which share 65–70% homology[8]. On the basis of genomic diversity, all the four DENV serotypes are further classified into different genotypes[9]. The intensity and impact of a dengue outbreak is influenced by a shift in the circulating DENV strains, ability of pre-existing immunity to counter the circulating viruses and the environmental factors such as temperature and rainfall which is linked to vector breeding[10]. Mutations in DENV proteins and/or viral evolution leading to emergence of novel clades, introduction of new genotypes into circulation have been associated with dengue outbreaks[11–14]. The live-attenuated tetravalent vaccine using the yellow fever virus backbone (CYD-TDV) was licensed for human use in few countries, however, the long term safety follow-up data showed increased risk of hospitalization in children aged less than 9 years of age[15]. This observation has led to WHO recommending pre-vaccination screening and vaccination of only seropositive individuals[16].

Studies on seroprevalence and characterization of neutralizing antibodies in the Indian population are limited. Although antibodies are presumed correlates of protection in dengue, there is no information on how pre-existing humoral immunity counters subsequent dengue infection during acute phase. Our objective was to identify association between neutralizing antibody profiles from prior exposure and viral RNA levels during subsequent infection. In a large serosurvey from 15 out of 30 states of India, the reported seroprevalance varied by age from 28.3% in 5–8 age group, 41% in the 9–17 age group to 56.2% in the 18–45 age group[17]. 72.5% of the samples tested for neutralization ($n = 500$) showed a multitypic antibody profile. In another study, the dengue seropositivity in Pune City was 21.6% among <3 years age group and 77.3% in 16–18 years. All the adults above >70 years of age were seropositive and 69.2% of the seropositive samples tested for neutralizing antibodies ($n = 119$) were capable of neutralizing all four serotypes of DENV[18]. Another community based study reported the seroprevalence in children aged 5–10 years ranging from 23.2% in Kalyani, West Bengal to 80.1% in Mumbai[19]. Therefore, the seroprevalence and the profile of antibodies vary by age, geographical region and between the urban and rural settings in India. In this study, we estimated the prevalence of dengue antibodies in leftover samples collected for diagnosis of suspected dengue cases during the dengue season of 2018–2019 from the National Capital Region of India which covers an area of approximately 50,000 Km2 including the National Capital Territory of Delhi and the three surrounding states which are hotspots for Dengue outbreaks. We quantified dengue RNAemia (DENV RNA genomic equivalents) and identified the serotype of the infecting virus. We isolated the dengue virus from the clinical samples and established a focus reduction neutralization titer (FRNT) assay which has a better throughput than the conventional plaque reduction neutralization titer (PRNT) assays. We estimated the neutralizing antibody titers using all the four serotypes of Indian isolates in a subset of samples. A further subset of samples were also tested to assess neutralization of international dengue isolates and also other flaviviruses namely, Zika virus, Japanese encephalitis virus and West Nile virus.

Our study is a comprehensive overview of binding and neutralizing antibodies for dengue viruses in a cohort of patients with acute febrile illness and shows correlation between pre-existing antibodies, RNAemia and circulating dengue viruses in the National Capital Region of India. We observe antibody cross-reactivity with related flaviviruses and a reduced ability of these antibodies to neutralize the parental strains which are part of live-attenuated dengue vaccine which are currently being evaluated for efficacy in other countries. Our results suggest a need to characterize baseline immune responses in the population to dengue as part of vaccine development and implementation strategies in India.

## Methods

**Ethics statement**. Discard samples from a diagnostic lab was used. No personal identifiers or clinical data of the patients were obtained with the samples. The Institutional Ethics Committee for Human Research of Translational Health Science and Technology Institute has exempted this study from review as anonymous, leftover samples were obtained from a diagnostic lab to generate baseline data and create reference panels to establish various assays for dengue vaccine development in the bioassay laboratory.

**Human samples**. Anonymous, discard samples ($n = 412$) post-dengue diagnosis by NS-1 or IgM ELISA were obtained from a centralized diagnostic lab in the National Capital Region (NCR) of Delhi which had received majority of the samples from hospitals of the NCR in New Delhi, Delhi, Gurugram, Faridabad, Ghaziabad, NOIDA for dengue diagnosis. Samples were transported from various collection centres across the NCR during September and October months of 2018 and 2019 to a central lab and stored at −20 °C till further transfer to the bioassay laboratory where the samples were stored at −80 °C till further testing in various assays. Only six serum samples were from 2020 which was used only in neutralization assays to determine cross-neutralization antibodies to flaviviruses and international isolates. We measured exposure to dengue by IgG and IgM ELISAs. Prior exposure to dengue was assessed by indirect ELISA and neutralizing antibody assays. Acute dengue infection was verified by detection and quantitation of dengue virus RNA by quantitative RT-PCR. The infecting virus serotype was determined by nested RT-PCR as described in the following sections. Two independent teams processed samples for ELISA and RNA isolations (for serotyping and viremia) and they were blinded for results of one another at the time of performing the assays.

**Cell lines**. LLCMK2 cells were obtained from European Collection of Authenticated Cell Cultures (ECACC – Cat no: 85062804). C6/36 insect cells and BHK-21 cells were obtained from American Type Culture Collection (ATCC) (Cat no. CRL-1660 and CCL-10 respectively). All cell lines were grown in Minimum Essential medium (MEM) (Gibco-11090-073) containing 10% heat-inactivated fetal bovine serum (FBS) (Gibco-16140-071), 100 U/mL of penicillin, streptomycin and Glutamine (Gibco-10378-016). LLCMK2 and BHK-21 were grown at 37 °C whereas C6/36 were grown at 28 °C in a humidified 5% $CO_2$ incubator.

**Viruses**. Dengue virus was isolated from serum of patients by infecting C6/36 cells as described previously[20]. We isolated DENV-1 through DENV-4 isolates from the clinical samples by passaging three times in C6/36 cells. Virus stocks were titered by

plaque assay on BHK-21 cells as described below. Different Serotypes of Dengue Virus, Dengue 1 (Genbank; ON799266), Dengue 2 (Genbank; ON799267), Dengue 3 (Genbank; ON799401) and Dengue 4 (Genbank: OP310810) were propagated in C6/36 cells and passaging was limited to four passages. International Dengue Isolates used in this study are the backbone recombinant viruses used to generate the live-attenuated vaccine rDENV1 WP (DEN1-Western Pacific Nauru), rDENV3-7164 (DEN3-Sleman/78) and rDEN4 7-4A-1A2 (DENV4-Dominica/81) obtained from National Institute of Health, USA[21–23]. For DENV-2, the New Guinea-C prototype was used. The Indian isolates of JEV and WNV used in this study have been reported earlier[24]. Zika virus (MR-766 strain) was obtained from BEI resources (NR-50065) and virus stocks were prepared in C6/36 cells. All the dengue clinical isolates were authenticated by whole genome sequencing as described in the supplementary methods section.

**Plaque assay**. Viral titers of all the isolates were determined by plaque assay on BHK-21 cells. Briefly, 50,000 BHK-21 cells were seeded in each well of a 24 well plate. Virus was serially diluted 10-fold in assay diluent containing Minimum Essential medium (MEM) (Gibco-11090-073) containing 2% heat-inactivated fetal bovine serum (FBS) (Gibco-16140-071), 100 U/mL of penicillin, streptomycin and Glutamine (Gibco-10378-016) and BHK-21 cells were infected with serial dilutions of virus at 37 °C, 5% $CO_2$ for 1 h on rocker. Post-infection inoculum was removed and cells were overlaid with 0.5% carboxymethylcellulose solution (CMC; Sigma – C4888) prepared in assay diluent as described above. The plates were incubated at 37 °C under a humidified atmosphere of 5% $CO_2$ for 76 h for DENV-1 and 2, for 90 h for DENV-3, 80 h for DENV-4 and 48 h for ZIKV. After infection for the required time CMC solution was aspirated and cells were fixed with 3.7% formaldehyde solution for 30 min. Formaldehyde solution was washed with tap water and cells were stained with crystal violet solution for 10 min. Plates were washed with tap water and allowed to dry. Post-drying plaques were counted and titers represented as Plaque Forming Unit (PFU)/ml.

**Enzyme-linked immunosorbent assay (ELISA)**. Panbio Dengue Early ELISA for NS1 antigen (Cat No. 01PE40), Panbio Dengue IgM capture ELISA (Cat No. 01PE20), Dengue IgG Capture ELISA (01PE10) and Panbio Dengue IgG Indirect ELISA (Cat No. 01PE30) was performed for the qualitative detection of NSI, IgM & IgG antibodies respectively as per the kit instructions. Samples with values above the cut-off values in indirect ELISA were declared as seropositive for dengue. Samples below the limit of detection (LOD) were assigned a values of LOD/2 which is 5.5 for NS1, IgM and Indirect IgG ELISA and 11 for Direct IgG ELISA.

**Quantitative RT-PCR for dengue RNA copy number estimation**. RNA was extracted from 50 µl serum samples using NucleoSpin RNA Isolation Kit by MACHEREY-NAGEL. (Cat.No.:740956). 4 µl of RNA was used in qRT-PCR reactions. The RT-PCR was performed by using SOLIScript 1-step probe kit (Cat. No. 08-57-00250)-with the following cycling conditions: 50 °C for 15 min for reverse transcription, 95 °C for 10 min for initial denaturation, and 40 cycles of 95 °C for 15 s and 60 °C for 1 min using Bio-Rad CFX-96 real-time PCR System. 200 nM concentration for primers and probes (Supplementary Table S1) were used. A portion of the 3' untranslated region of the DENV genome starting from 10401-10702 was amplified and cloned into pGEM®-T-Easy vector (Promega). This clone was linearized using Sac II enzyme and in vitro transcribed using the SP6 RNA

polymerase (Promega, Cat no: P1280). The transcript was purified and used as a template for generating standard curve to estimate the copy number. The limit of detection of the assay is 2230 copies/ml of serum. Samples below the limit of detection was assigned a LOD/2 value of 1115 copies/ml of serum.

**DENV serotyping**. Dengue serotyping was performed by reverse transcription nested PCR as described earlier[25] with slight modification and the primers details are mentioned in (Supplementary Table S2). Briefly the reaction mixture contains 11.3 µl of nuclease free water, 5 µl of M-MLV RT 5X buffer (Promega, cat no: M5313), 1.5 µl of 25 mM $MgCl_2$ (NEB, cat no: B9021S), 0.5 µl of 10 mM dNTPs (Promega, cat no: U1330), 1.25 µl of 0.1 M DTT (Sigma, cat no: 43816), 0.5 µl of 10 µM D1 and D2 Primers, 0.25 µl of MMLV Reverse Transcriptase (200 U/µl) (Promega, cat no: M1705) and 0.2 µl of Taq DNA Polymerase (5 U/µl) (NEB, cat no: M0273L). 4 µl of extracted RNA was used as a template in a 25 µl reaction volume for first step PCR. Reactions were incubated in a thermal cycler (ABI Veriti) for 60 min at 42 °C, followed by 94 °C for 2 min, 35 cycles of 94 °C for 30 s, 55 °C for 1 min, and 72 °C for 1 min, and a final extension at 72 °C for 5 min. Nested PCR was carried out with 1.25 µl of diluted (1:50) material from first PCR and used as a template for the second PCR in a 25 µl reaction volume containing 16.11 µl of nuclease-free water, 2.5 µl of Standard Taq buffer (Mg-free) (NEB, cat no: B9015S), 1 µl of 25 mM $MgCl_2$ (NEB, cat no: B9021S), 0.5 µl of 10 mM dNTPs (Promega, cat no: U1330), 1 µl of 10 µM D1, 1 µl 10 µM TS1, 0.5 µl 10 µM TS2, 0.315 µl 10 µM TS3 and 0.625 µl 10 µM TS4 Primers and 0.2 µl of Taq DNA Polymerase (5 U/µl) (NEB, cat no: M0273L). PCR reaction was performed at 94 °C for 3 min, followed by 25 cycles of 94 °C for 30 s, 52 °C for 1 min and 72 °C for 1 min, with a final extension at 72 °C for 5 min. Amplicon was resolved using Agarose Gel electrophoresis or Qiagen QIAxcel capillary electrophoresis system to visualize the amplicon size and intensity. A sample containing DENV-1, 2, 3 or 4 was identified by size 482,119, 290, or 392 ± 5 % base pairs (bp), respectively. Whole genome amplification and sequencing of DENV isolates has been described in detail under Supplementary methods. The list of primers used in generation of amplicons for whole genome sequencing is provided in Supplementary data 1.

**Microneutralization (MNA) assays**. Dengue MNA assay was established based on our previously reported protocol for SARS-CoV-2 neutralization assay[26] with modifications as required for each of the flaviviruses. LLCMK2 cells were seeded at a density of 20,000 cells/well respectively in wells of 96-well plates. Serum samples were heat-inactivated at 56 °C for 30 min and were serially diluted (2-fold dilutions with 1:25 as starting dilution in the tube for DENV and ZIKV and 1:10 for JEV and WNV) in a 96 well plate in assay diluent (Minimum Essential Medium (MEM) (Gibco, Cat No.11090073) supplemented with 2% heat-inactivated Fetal Bovine Serum (FBS) (Gibco, Cat no. 16140-071), 1X Penicillin-Streptomycin-L-glutamine solution (PSG) (Gibco Cat no. 10378016). Serum dilutions were mixed 1:1 with diluted virus stocks (30 µl serum dilution mixed with 30 µl virus dilution) making the final starting dilution at 1:50 for DENV and ZIKV and 1:20 for JEV and WNV. The virus dilution was pre-optimized to give final foci count of 30–150 foci per well. Wells with diluted virus mixed with media alone served as virus control wells while wells without virus served as cell control wells. Plates containing serum-virus complexes were incubated at 37 °C for 60 ± 5 min for neutralization. Post-neutralization, LLCMK2 cells were washed once with 1X Phosphate Buffer Saline (PBS) and 40 µl serum-virus complex was added from neutralization plate to respective

wells of the 96-well LLCMK2 cell plate. After incubating the plate at 37 °C for 60 ± 5 min, serum-virus complex was removed and cells were overlaid with 100 μl 0.75% carboxymethylcellulose solution (CMC; Sigma – C4888) prepared in assay diluent for DENV-1, 2 and 3 and 1 % CMC solution for DENV-4, 2% CMC solution for ZIKV and 1% CMC for JEV and WNV. The plates were incubated at 37 °C under a humidified atmosphere of 5% $CO_2$ for 70 ± 2 h for DENV-1, 2 and 3, 44 ± 2 h for DENV-4, 22 ± 1 h for WNV, JEV and 51 ± 1 h for ZIKV. Post-incubation, CMC solution was removed from the wells and cells were fixed with 4% paraformaldehyde (Sigma-P6148) solution in 1X PBS (pH 7.4) for 20 min at room temperature (RT). After fixing, cells were washed two to three times with PBS and were permeabilized using permeabilization buffer [20 mM HEPES, pH 7.5 (HiMedia-MB016), 0.1% Triton-X-100 (Qualigens-Q10655), 150 mM sodium chloride (HiMedia-MB023), 5 mM EDTA (Sigma-E5134)] for 10 min at RT. Cells were then incubated with pan-flavivirus 4G2 anti-E antibody [hybridoma supernatant diluted 1:10 in permeabilization buffer] for 1 h at RT. For DENV plates, cells were washed thrice with permeabilization buffer and HRP-tagged anti-mouse IgG-secondary antibody (Invitrogen-A16072) (diluted 1:500 in permeabilization buffer) was added to each well for 1 h at RT. Wells were washed twice with permeabilization buffer and once with 1X PBS followed by addition of True-Blue peroxidase substrate (KPL Immunoassay-55100030) to each well and incubated for 10 min at RT. Reaction was terminated by washing the wells with 1X PBS followed by decanting and drying the wells. Microplaques developed after staining were acquired AID iSPOT reader (AID GmbH, Strassberg, Germany) using LED light and selecting the LED Channel. For ZIKV, JEV and WNV plates, cells were washed three times with permeabilization buffer and goat anti-mouse Alexa 488 secondary antibody (Jackson; Cat No. 115545003) (diluted 1:500 in permeabilization buffer) was added to each well for 1 h at RT on rocker followed by washing thrice with MilliQ water and 50 μl 1X PBS was added to plates then were acquired after removing PBS from wells in AID iSPOT reader (AID GmbH, Strassberg, Germany) using Xenon lamp and selecting the FITC Channel. AID EliSpot 8.0 software was used for acquisition. Neutralization titers were calculated in reference to foci counts in virus control wells. The raw data generated from the AID iSpot analyser in a 96-well format was pasted in apre-defined protocol template for calculation of $FRNT_{50}$ by using $log_{10}$ transformed dilution value and neutralization percentages in an XY format. The Point-to-Point curve fit using a linear equation to fit each pair of data points was used to calculate the FRNT50 value using SoftMax Pro GxP Software v7.1.1 (Molecular Devices).

**Statistics and reproducibility**. The subgroup analyses were based on age, seropositivity, viral RNA levels and infecting dengue virus serotype. All statistical analysis and graphical representations were performed using GraphPad Prism Software version 9 and above (GraphPad Software Inc, San Diego, USA). Statistical analysis of estimated using non-parametric tests as indicated in the respective figure legends. A P-value of >0.05 was considered significant. Sample numbers indicated as "n" in each figure and figure panels indicate independent biological samples tested in two technical replicates and mean value of each sample was used for analysis and graphical representation.

**Reporting summary**. Further information on research design is available in the Nature Portfolio Reporting Summary linked to this article.

## Results

**Dengue seropositivity increases by age**. The median age of subjects was 30 years (Range 0–85 years). 73 (18%) participants were in the 0–15 year group, 259 (63%) subjects were in the 16–45 years group and 80 (19%) were >45 years of age. Overall, 376 out of 412 samples were confirmed as dengue positive by either NS-1 or IgM ELISA by the diagnostic lab (Fig. 1a). Out of 412 samples tested for dengue NS-1, 318 samples (77.2%) were positive. 333 samples out of 412 had IgM ELISA results. 121 samples (36.3%) were positive for IgM (Fig. 1a). 36 samples were negative in both NS-1 and IgM ELISA and were considered as dengue negative. 114 samples were tested by the diagnostic lab with direct IgG ELISA, out of which 30 were positive and 3 samples had equivocal results (Fig. 1a). As most of the samples were from adults, we speculated that most of these could be secondary infections as about half of the adult population in urban India was shown to be seropositive by other studies[17, 18]. Therefore, we tested all the samples for the presence of dengue IgG by indirect ELISA which detects low levels of antibodies in samples in endemic settings. Results showed that 296 of the 412 samples (71.8%) were IgG positive by indirect ELISA. Twenty three out of 36 dengue-negative samples (NS-1 and IgM ELISA negative) were also positive by indirect ELISA indicating past exposure. We performed paired analysis of 30 samples that were positive by both the direct and indirect ELISA. As expected, indirect ELISA units were significantly higher than the direct ELISA confirming higher sensitivity of the assay (Fig. 1b). Three samples which had equivocal results by direct ELISA were positive in indirect ELISA. Out of the 81 samples that were negative by direct ELISA, 54 samples were positive by indirect ELISA (Fig. 1c) further confirming the utility of indirect ELISA in seroprevalence estimations in endemic settings and also for appropriate classification of primary and secondary dengue infections. The seropositivity across all age groups was estimated to be 72% (Fig. 1d) and as expected, dengue seropositivity increased with age with 52% (38 out of 73) in 0–15 years age group (Fig. 1e), 72.2% (187 out of 259) in 16–45 years group (Figs. 1f), 88.8% (71 out of 80) in >45 years group (Fig. 1g).

**Viremia negatively correlates with pre-existing antibody levels**. We next isolated total RNA from all the serum samples ($n = 412$) and DENV RNA levels were measured by quantitative RT-PCR as described previously[27]. 93·9% ($n = 387$) of the samples were positive for DENV RNA (Fig. 2a). We estimated correlation between viral RNA levels in serum with the IgG units of indirect ELISA (Panbio Units) in samples that were IgG positive without any detectable IgM ($n = 194$). We observed a significant, negative correlation between the IgG levels and dengue genome copy numbers suggesting a role for pre-existing IgG in restricting virus replication in these infected individuals (Fig. 2b). We found no correlation between IgM levels (Panbio Units) and viremia (Fig. 2c) in samples where only IgM was present ($n = 18$) which were all primary infections. However, the sample size is too small to draw any statistical inference for this lack of correlation. It is well known that the sub-neutralizing antibodies also cause antibody-dependent enhancement (ADE) of dengue infection in secondary infections. Therefore, we next categorized the samples into primary and secondary infections based on the presence or absence of IgG or IgM or both to analyse viral RNA levels in these sub-groups. Samples with no IgG or IgM (seronegative) or samples with IgM only and also those with IgM:IgG > 1·8 were classified as primary infections. Samples with IgG only or IgM:IgG < 1.8 were classified as secondary infections[28]. Contrary to previous studies, we observed a significantly higher viremia in primary infections (geometric mean of 2·76E + 06 $log_{10}$ genome

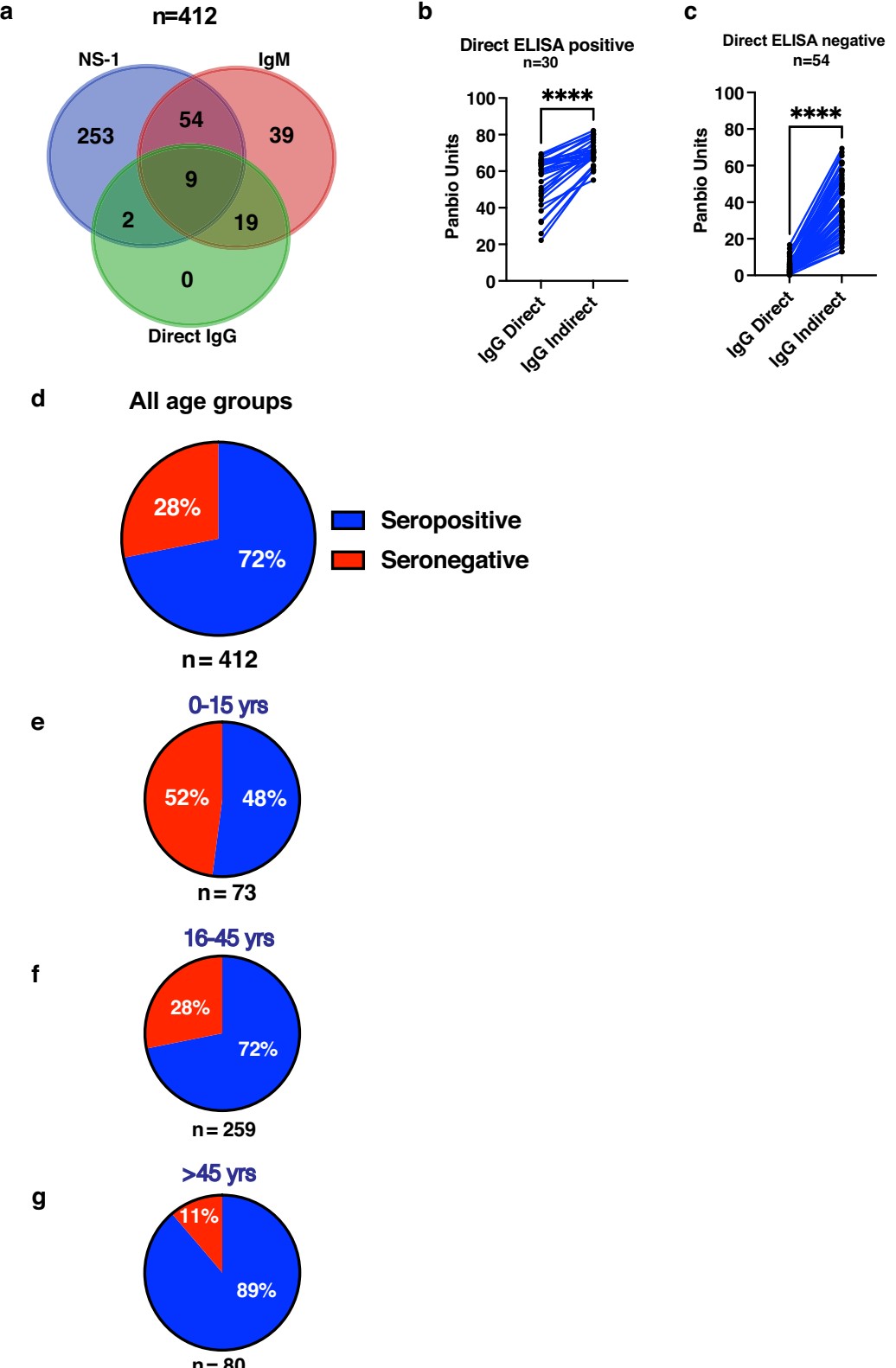

**Fig. 1 Dengue seropositivity during acute phase of dengue fever. a** Overview of dengue diagnostic data of the samples that were tested to be positive by either NS-1 (blue circle) or IgM (red circle) or direct IgG ELISA (green circle) (*n* = 412 independent samples). **b** 30 samples positive in direct IgG ELISA and, **c** 54 samples negative in direct IgG ELISA (IgG Direct) were compared with values obtained from indirect ELISA (IgG Indirect). Paired analyses was performed to compare the values. Significance between the observations were determined by Wilcoxon matched-pairs signed rank test. Two-tailed *P*-value is shown. ****$P < 0.0001$. Dengue antibody positivity was determined by indirect ELISA and the seropositivity was further segregated based on the age of the subjects. Seropositivity is represented by blue and seronegativity by red colour, (**d**) all age groups, **e** 0–15 years, **f** 16–45 years, **g** >45 years. Total number of samples "n" is indicated below the pie chart for each of the panels.

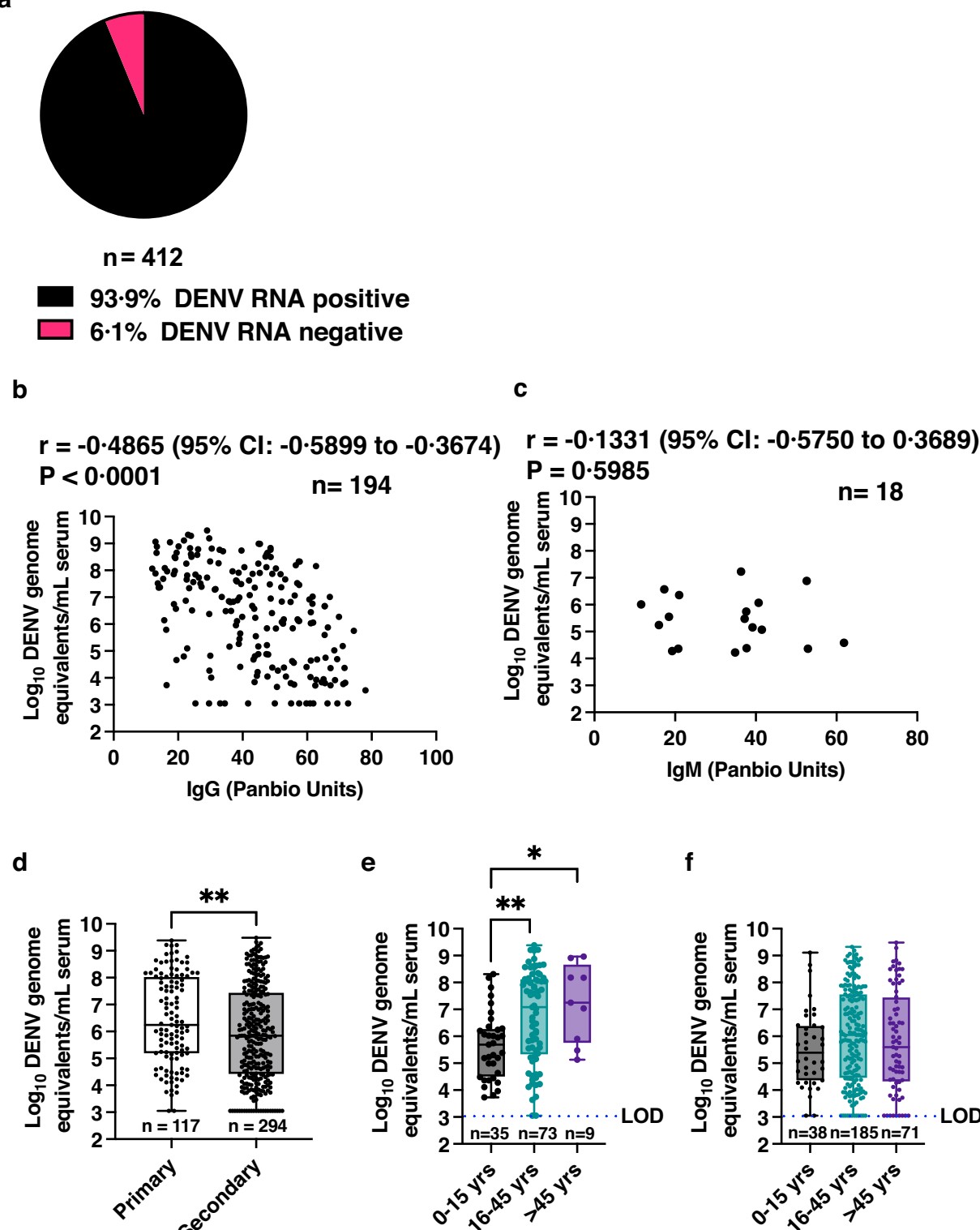

equivalents with 95% CI: 1·37E + 06 to 5·54E + 06) as compared to secondary infections (geometric mean of 8·10E + 05 $\log_{10}$ genome equivalents with 95% CI:5·07E + 05 to 1·29E + 06) (Fig. 2d). Thus, our data suggests that antibody levels show a negative correlation with viral RNA levels and primary dengue infections, where low or no IgG was present, are associated with higher viremia. These results are consistent with a previous observation which showed that the resolution of viremia is longer in primary infections as compared to secondary dengue[29]. We further analysed DENV RNA copy numbers in primary and secondary infections as per the three age groups categorized as described earlier. In primary dengue samples, the viral RNA levels were significantly higher in both the adult age groups as compared to paediatric samples (Table 1 and Fig. 2e) whereas in secondary dengue samples, the viral RNA levels were similar between all the three age groups (Table 1 and Fig. 2f).

**Fig. 2 Correlation between dengue viremia and antibodies. a** Total RNA was isolated from serum samples obtained from patients with acute febrile illness. DENV RNA levels were detected and quantitated by RT-PCR. Proportion of samples which were positive for viral RNA is shown in black and negative samples are indicated in pink. **b** Association between DENV RNA levels and IgG or (**c**) IgM antibody units as estimated by ELISA was determined by Spearman correlation analysis (r). Two-tailed P value is indicated. **** P < 0.0001. "n" indicates number of independent samples. **d** Levels of dengue genomic equivalents was segregated based on primary (clear box) or secondary dengue (grey box) infection. Two-tailed *P*-value was estimated by non-parametric, Mann-Whitney test. **P = 0.0063. **e, f** Levels of dengue genomic equivalents was segregated based on age groups (0–15 years (grey box), 16–45 years (green box) and > 45 years (purple box) as indicated from Primary and Secondary dengue infections respectively. Mean rank of each column was compared with mean rank of every other column using non-parametric Kruskal-Wallis test. Correction for multiple comparisons was applied by Dunn's multiple comparison test. **P = 0.0044, *P = 0.0285. Error bars in (**d–f**) represent Min and Max with range. LOQ is limit of quantitation of the assay. "n" in each of the panels indicates biologically independent samples. *P*-values are indicated only for observations where the differences are statistically significant.

**Table 1 DENV genome equivalents per mL of serum in patients with indicated age groups.**

| Primary/Secondary | Primary | | | Secondary | | |
|---|---|---|---|---|---|---|
| Age groups | 0–15 years | 16–45 years | >45 years | 0–15 years | 16–45 years | >45 years |
| Number of values (*n*) | 35 | 73 | 9 | 38 | 185 | 71 |
| Geometric mean | 4·43E + 05 | 5·32E + 06 | 1·67E + 07 | 3·45E + 05 | 1·07E + 06 | 6·21E + 05 |
| Lower 95% CI of geo. mean | 1·67E + 05 | 2·10E + 06 | 1·27E + 06 | 1·15E + 05 | 5·91E + 05 | 2·20E + 05 |
| Upper 95% CI of geo. mean | 1·18E + 06 | 1·34E + 07 | 2·20E + 08 | 1·03E + 06 | 1·93E + 06 | 1·75E + 06 |

**Serotype dominance may be influenced by pre-existing antibodies.** DENV-2 was the predominant infecting serotype in the New Delhi region from 2011 to 2015 with DENV-1 as the next dominant serotype and minimal to no cases of DENV-3 and DENV-4[20,27, 30, 31]. However, DENV-3 was identified as the predominant serotype in 2016 and 2017[30, 32]. To further monitor the infecting serotype during our study period, total RNA from all the 412 samples were used to identify the infecting DENV serotype by nested PCR[25] as this technique was likely to detect and amplify viral RNA for serotyping in samples which were negative in RT-PCR. Of the 412 samples, dengue virus serotype could not be determined in 140 samples, forty five were DENV-1, twenty seven were DENV-2, one hundred sixty nine were DENV-3 and twenty eight samples were positive for DENV-4 serotype. Three samples were positive for both DENV-1 and DENV-3 (Fig. 3a). 16 of the RT-PCR negative samples were serotype positive and 11 of these samples were secondary DENV-4 infections. We could not detect serotype in 140 samples most likely due to significantly lower amounts of viral RNA present in these samples (Supplementary Fig. S1). We further segregated the distribution of serotypes based on the year of sample collection. Of the 339 samples collected in 2018, serotype could not be determined for 128 samples (38%). Of the remaining samples, 48% were identified as DENV-3 serotype (Fig. 3b). Similarly, out of the 72 samples collected in 2019, twelve samples (16%) were indeterminate in nested PCR for serotyping. Of the remaining samples where serotype could be determined, 43% of the samples were positive for DENV-1 serotype (Fig. 3c). Thus our serotyping data shows a shift in circulating serotype from DENV-2 observed during 2011 to 2015 to DENV-3 as the predominant serotype in the years 2017 and 2018 and a further shift to DENV-1 in 2019 in the study region. We further analysed the DENV copy numbers based on the infecting serotype to assess if there are any serotype-dependent differences. We observed no significant difference in the viral RNA levels between DENV-1, DENV-2 and DENV-3. However, DENV-4 infected samples had significantly lower levels of viral RNA as compared to the other three serotypes (Fig. 3d and Table 2). As expected, in samples where the infecting serotype could not be ascertained also had significantly lower levels of viral RNA (Fig. 3d). These results suggest that patients infected with DENV-4 may have had lower viremia as compared to the other three serotypes.

**Neutralizing antibodies as a potential driver of circulating virus selection.** With over half of the adult population seropositive for dengue in India, the cross-talk between the pre-existing humoral immunity and infecting virus is likely to play a major determining role in the emergence of circulating dengue viruses that can either evade antibody response or are more replication competent. We speculated that the level of neutralizing antibodies, either homotypic or heterotypic, may determine the susceptibility to infection with circulating DENV serotype. Individuals with antibodies to any of the serotypes may be protected from reinfection either due to homotypic protection or cross-protection unless the antibody levels are below the level required for protection or the infecting serotype escapes pre-existing humoral immunity. We isolated DENV-1-4 isolates from samples that were positive for DENV RNA. These Indian isolates were used to establish a 96-well plate-based focus-reduction neutralization assay (FRNT) using pan-flavivirus envelope antibody, 4G2, for staining the infectious foci (Supplementary Fig. S2A). We were able to perform FRNT assays on 215 samples where sufficient volume of samples were available for testing all the four DENV serotypes. 35 dengue-negative samples which had no neutralizing antibodies for any of the four serotypes was used to set the limit of quantitation (LOQ) of the assay which was set to a $FRNT_{50}$ of 50 (50% neutralization of virus at 1:50 dilution of the serum). All samples negative in FRNT assay and with a $FRNT_{50}$ value of less than 50 were assigned a value of 25 (LOQ/2) (Supplementary Fig. S2B). 31 samples had monotypic antibodies to any one of the serotypes (Eight for DENV-1; Nine for DENV-2; Twelve for DENV-3; Two for DENV-4) (Fig. 4a) and there were nine DENV-1, five DENV-2, eleven DENV-3, one DENV-4 and five serotype indeterminate infections in this group of 31 patients with monotypic neutralizing antibodies. 149 samples had neutralizing antibodies to more than one serotype (Fig. 4b). Consistent with the predominantly DENV-2 infection in the study region during the previous years, the neutralizing antibody titers for DENV-2 was the highest with the antibody titers of DENV-2 > DENV-1 > DENV-3 > DENV-4 (Table 3 and Fig. 4b). The lower levels of $FRNT_{50}$ titers observed for DENV-3 and DENV-4 coincided with DENV-3 and DENV-4 being the predominant infecting serotype in these subjects (Fig. 4c). In about 36% of these samples, the serotype of the infecting virus could not be determined and most likely due to low viral RNA levels as shown earlier (Fig. 3d). To further verify that the neutralizing

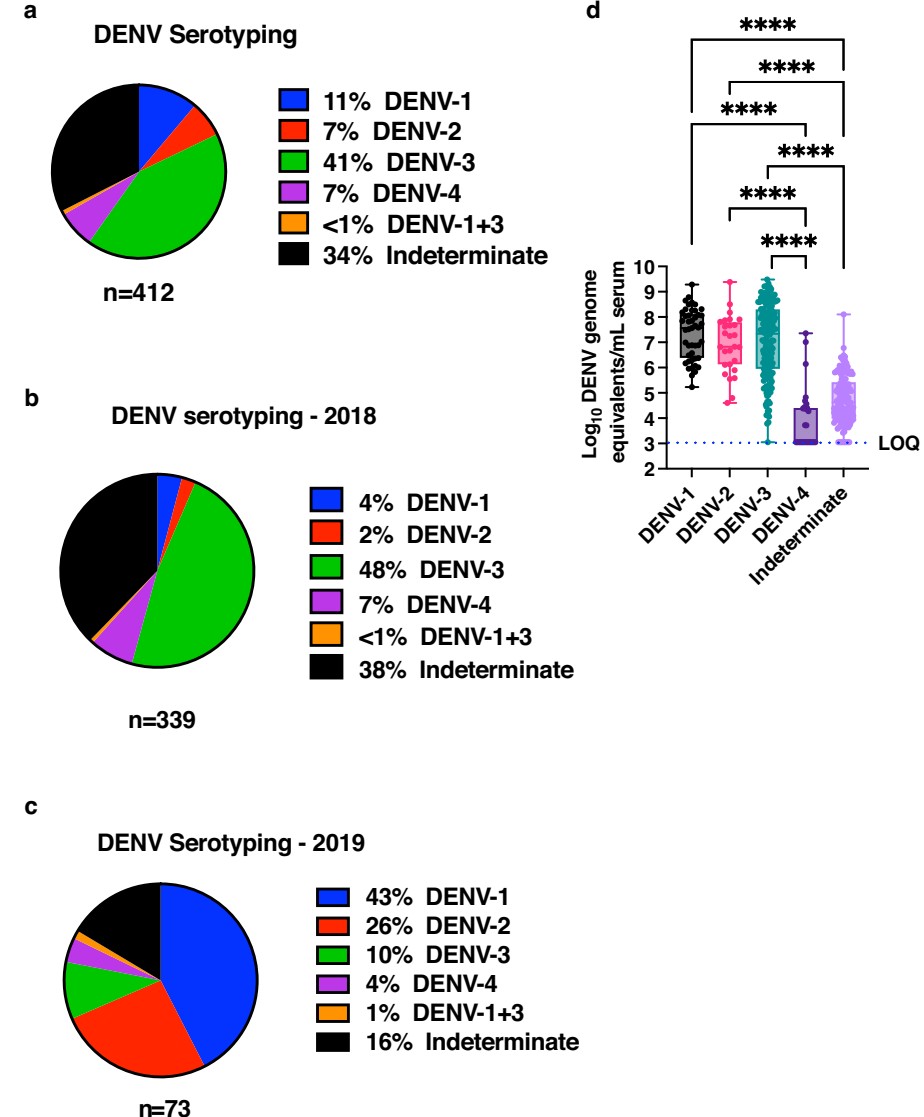

**Fig. 3 Co-circulation of dengue serotypes and serotype dominance. a** Proportion of samples positive for infecting dengue serotype as determined by nested PCR using total RNA isolated from serum samples is represented by the colours as labelled in the figure: Blue – DENV-1, red-DENV-2, green – DENV-3, purple – DENV4, orange - (DENV-1 + DENV-3) and black - indeterminate serotype. The segregation of serotyping data based on the year of sample collection is shown for (**b**) 2018 and (**c**) 2019. **d** DENV RNA levels in samples segregated as per the infecting serotype is shown. DENV-1: grey, DENV-2:pink, DENV-3:green, DENV-4:dark purple, indeterminate: light purple. Mean rank of each column was compared with mean rank of every other column using non-parametric Kruskal-Wallis test. Correction for multiple comparisons was applied by Dunn's multiple comparison test. ****$P < 0.0001$. Error bars in (**d**) represent Min and Max with range. LOQ is limit of quantitation of the assay. "n" in each of the panels indicates biologically independent samples. $P$-values are indicated only for observations where the differences are statistically significant.

**Table 2 DENV genome equivalents as per the infecting serotype.**

| Infecting virus | DENV-1 | DENV-2 | DENV-3 | DENV-4 | Indeterminate |
|---|---|---|---|---|---|
| Number of values (n)* | 45 | 27 | 169 | 28 | 140 |
| Geometric mean | 2·12E + 07 | 8·33E + 06 | 1·16E + 07 | 8·20E + 03 | 5·24E + 04 |
| Lower 95% CI of geo. mean | 1·06E + 07 | 2·95E + 06 | 6·92E + 06 | 2·74E + 03 | 3·60E + 04 |
| Upper 95% CI of geo. mean | 4·23E + 07 | 2·35E + 07 | 1·93E + 07 | 2·45E + 04 | 7·60E + 04 |

*Three samples were positive for dual serotypes (DENV-1 and DENV-3) and were excluded.
*DENV-1* Dengue virus-1, *DENV-2* Dengue virus-2, *DENV-3* Dengue virus-3, *DENV-4* Dengue virus-4.

antibody levels may influence susceptibility to the circulating serotype and viral replication, we segregated the multitypic antibody samples based on the infecting DENV serotype and in the group where serotype could not be determined. As expected,

DENV-2 antibody levels were highest in all the five groups and DENV-1 antibody levels were not significantly different compared to DENV-2 in these groups (Fig. 4d–h). DENV-3 antibody levels were significantly lower compared to DENV-2 levels only

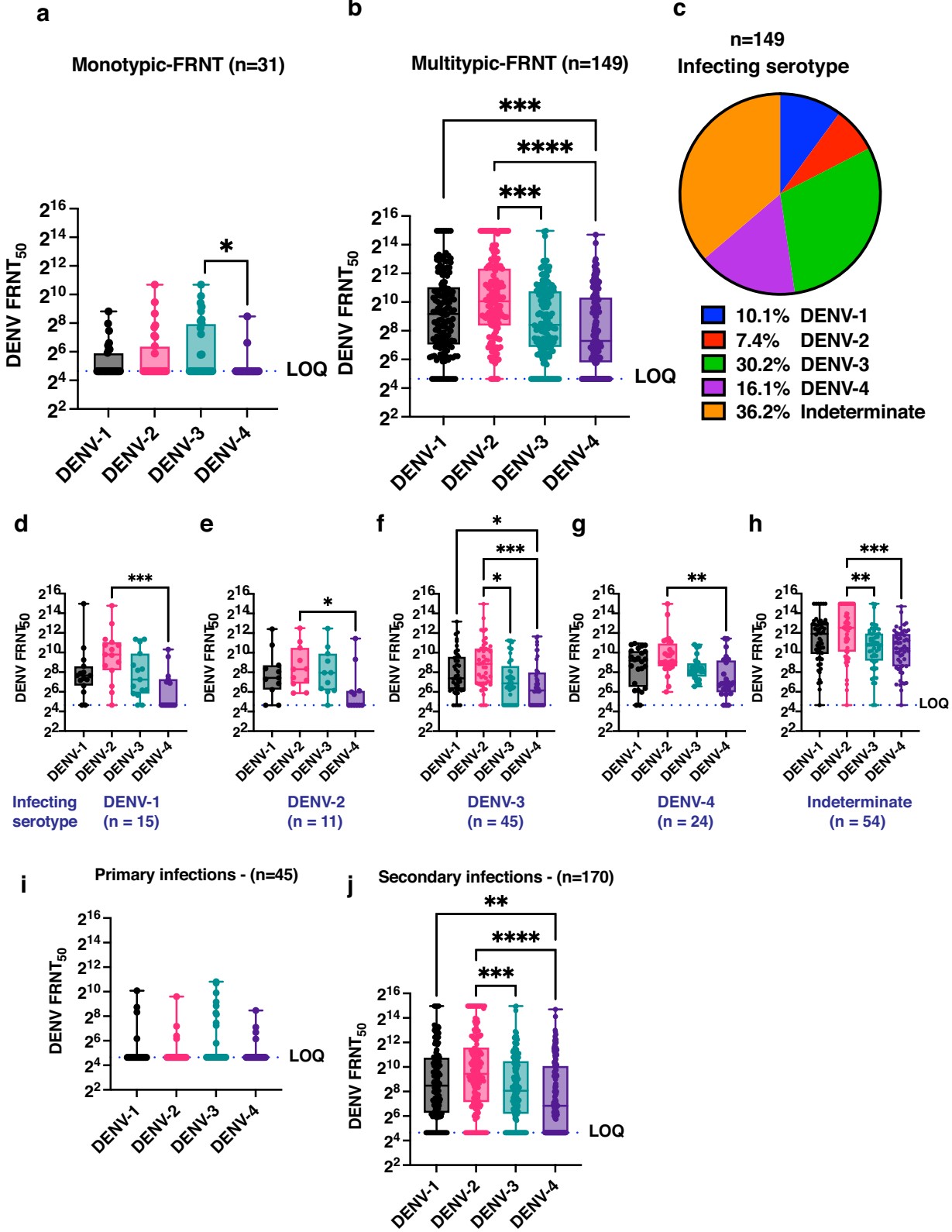

in samples where the infecting serotype was either DENV-3 or the serotype indeterminate samples (Fig. 4f, 4h). The samples with indeterminate serotype had the highest level of neutralizing antibodies further strengthening the argument for a direct negative correlation between neutralizing antibody levels and viral replication (Fig. 4h). As we observed a significantly higher viremia in primary dengue infections (Fig. 2d), we next

segregated FRNT data from 215 samples as per the primary ($n = 45$) or secondary infection ($n = 170$) status. Thirty two of the 45 primary dengue samples showed no neutralizing antibodies to any of the four serotypes. Of the remaining thirteen samples, ten samples had antibodies to any one of the DENV serotype and three samples had multitypic neutralizing antibody response (Fig. 4i). This is consistent with previous studies which

**Fig. 4 Association of neutralizing antibody response with infecting DENV serotype.** The four DENV serotypes are represented by the following colours in all the figure panels depicting FRNT data: DENV-1:grey, DENV-2:pink, DENV-3:green, DENV-4:dark purple. **a** Neutralizing antibody titers against any one of the four serotypes of DENV, as estimated by FRNT assay in a subset of samples ($n = 31$). **b** Neutralizing antibody titers against two or more serotypes of DENV as estimated by FRNT assay in a subset of samples ($n = 149$). Mean rank of each column was compared with mean rank of every other column using non-parametric Kruskal-Wallis test. Correction for multiple comparisons was applied by Dunn's multiple comparison test. *$P = 0.0213$; ***$P = 0.0002$; ****$P < 0.0001$. **c** Distribution of infecting dengue serotypes from the samples showing multitypic neutralizing antibody response represented by the colours: blue – DENV-1, red-DENV-2, green – DENV-3, purple – DENV4, orange - indeterminate serotype). **d–h** $FRNT_{50}$ values of samples classified based on the infecting DENV serotype or serotype indeterminate samples. Mean rank of each column was compared with mean rank of every other column using non-parametric Kruskal-Wallis test. Correction for multiple comparisons was applied by Dunn's multiple comparison test.***$P = 0.0009$ in (**d**). *$P = 0.0483$ in (**e**). *$P = 0.0443$ for DENV-1 vs DENV-4 and $P = 0.0101$ for DENV2- vs DENV-3 in (**f**). $P = 0.0012$ in (**g**). $P = 0.0030$ for DENV-2 vs DENV-3 and $P = 0.0002$ for DENV-2 vs DENV-4 in (**h**). LOQ is limit of quantitation of the assay. **i, j** $FRNT_{50}$ values for all the four dengue virus serotypes in samples classified as primary or secondary dengue infection respectively. Mean rank of each column was compared with mean rank of every other column using non-parametric Kruskal-Wallis test. Correction for multiple comparisons was applied by Dunn's multiple comparison test. ns: not significant. **$P = 0.0020$; ***$P = 0.0006$; ****$P < 0.0001$. "n" in each of the panels indicates biologically independent samples. Error bars represent Min and Max with range. $P$-values are indicated only for observations where the differences are statistically significant.

---

**Table 3 DENV $FRNT_{50}$ titers in serum of dengue patients ($n = 149$).**

| Serotype of virus used | DENV-1 | DENV-2 | DENV-3 | DENV-4 |
|---|---|---|---|---|
| Geometric mean (95% CI) | 605 (444–823) | 1147 (837–1571) | 410 (310–543) | 250 (184–338) |

*DENV-1* Dengue virus-1, *DENV-2* Dengue virus-2, *DENV-3* Dengue virus-3, *DENV-4* Dengue virus-4.

---

have reported multitypic neutralizing antibodies in early convalescent sera of primary dengue patients[33–35]. Out of the 170 secondary infection samples, three samples showed no detectable neutralizing antibodies to any of the serotypes despite being positive in indirect IgG ELISA. Twenty one sample had monotypic neutralizing antibodies and 146 samples had neutralizing antibodies to more than one serotype (Fig. 4j). Thus the lower levels of neutralizing antibodies in primary infections correlate with higher viremia.

**Antibody cross-reactivity with other flaviviruses.** Flavivirus infections have been shown to generate antibodies that are capable of neutralizing or enhancing infection of antigenically-related flaviviruses[36]. We next tested a subset of samples ($n = 89$) for neutralizing antibodies to some of the mosquito-borne flaviviruses circulating in India namely, Zika virus (ZIKV), Japanese encephalitis virus (JEV) and West Nile virus (WNV)[37,38]. Based on 13 (out of 89) samples that were tested negative for all the seven flaviviruses, we set a cut-off of $FRNT_{50}$ of 50 for ZIKV and $FRNT_{50}$ of 20 for JEV and WNV. Pairwise comparison of samples for neutralizing antibody titers showed a significantly lower $FRNT_{50}$ values for ZIKV which is closely-related to DENV but belongs to the Spondweni antigenic serogroup within the flavivirus genus[39]. There was minimal to no cross-reactivity to JEV and WNV which are part of a separate serogroup (Supplementary Table S3). The geometric mean of $FRNT_{50}$ values for ZIKV was 90 (62–132) which is slightly higher than JEV 26 (95% CI: 19–35) and WNV 53 (95% CI: 37–77). This also suggests lack of exposure or minimal exposure to ZIKV, JEV and WNV in the study region which is supported by previous reports which have shown localized pockets of circulation of these viruses in India[37, 38]. To further understand the extent of cross-reactivity in samples to one or more of the flaviviruses, we further segregated the samples based on the $FRNT_{50}$ values for different combination of flaviviruses. 12 samples showed neutralizing antibodies to only DENV (one or more serotypes) and as shown earlier, the

$FRNT_{50}$ values for DENV-4 was the lowest compared to the other three serotypes (Fig. 5a and Supplementary Table S4). 6 samples had neutralizing antibodies to DENV and ZIKV but not to WNV and JEV (Fig. 5b and Supplementary Table S5). 8 samples had antibodies to DENV and WNV but not for ZIKV and JEV, however, the levels of WNV neutralizing antibodies were significantly lower as compared to DENV-1-3 antibodies (Fig. 5c and Supplementary Table S6). 16 samples had neutralizing antibodies to DENV, ZIKV and WNV but not JEV and the level of $FRNT_{50}$ values for WNV was significantly lower compared to DENV-1-3 (Fig. 5d and Supplementary Table S7). 14 samples were positive for DENV, JEV and WNV but not ZIKV and the $FRNT_{50}$ values between these flaviviruses were not significantly different (Fig. 5e) and Supplementary Table S8). 13 samples had antibodies to all the seven flaviviruses tested (Fig. 5f and Supplementary Table S9). No sample was exclusively positive for ZIKV or WNV in FRNT assays. 4 samples were positive in only JEV $FRNT_{50}$ assays with very low levels of JEV-neutralizing antibodies ($FRNT_{50}$ of 80, 33, 24 and 22 respectively). One sample was positive for both DENV and JEV ($FRNT_{50}$ of 117, 217, 77 and 66 for DENV-1-4 and $FRNT_{50}$ of 24 for JEV respectively). 2 samples were positive for ZIKV neutralizing antibodies ($FRNT_{50}$ of 58 and >3200) and this sample was positive in JEV FRNT assay with values just above the limit of quantitation ($FRNT_{50}$ of 28 and 29). Overall, 37 of the 76 DENV FRNT positive samples had antibodies to ZIKV, 34 of the 76 samples had antibodies to JEV and 51 of the 76 samples were positive for WNV antibodies in FRNT assays suggesting circulation of one or more flaviviruses in the population leading to elicitation of flavivirus-cross reactive antibodies.

**Reduced ability to neutralize international DENV isolates.** Currently, there are no licensed vaccines for Dengue in India and the live-attenuated vaccine developed by National Institute of Health, has been licensed to Indian manufacturers for which Phase II/III clinical trials are being planned. Recent data from COVID-19 pandemic has shown how changes in the antigenic sites leads to emergence of variants of concern that escape neutralization by antibodies. Whole genome sequencing and further analysis indicated that all the four Indian isolates of dengue virus used in our neutralization experiments have evolved into separate genotypes relative to the international isolates used in this study (Supplementary Fig. S3 to S6) which is in agreement with our and other previous reports from India[40–42]. The international DENV-1 isolate belonged to genotype IV while the Indian isolate used in our study was from genotype III along with other Indian isolates isolated within the last ten years (Supplementary Fig. S3). The cosmopolitan genotype of DENV-2 has been circulating in India

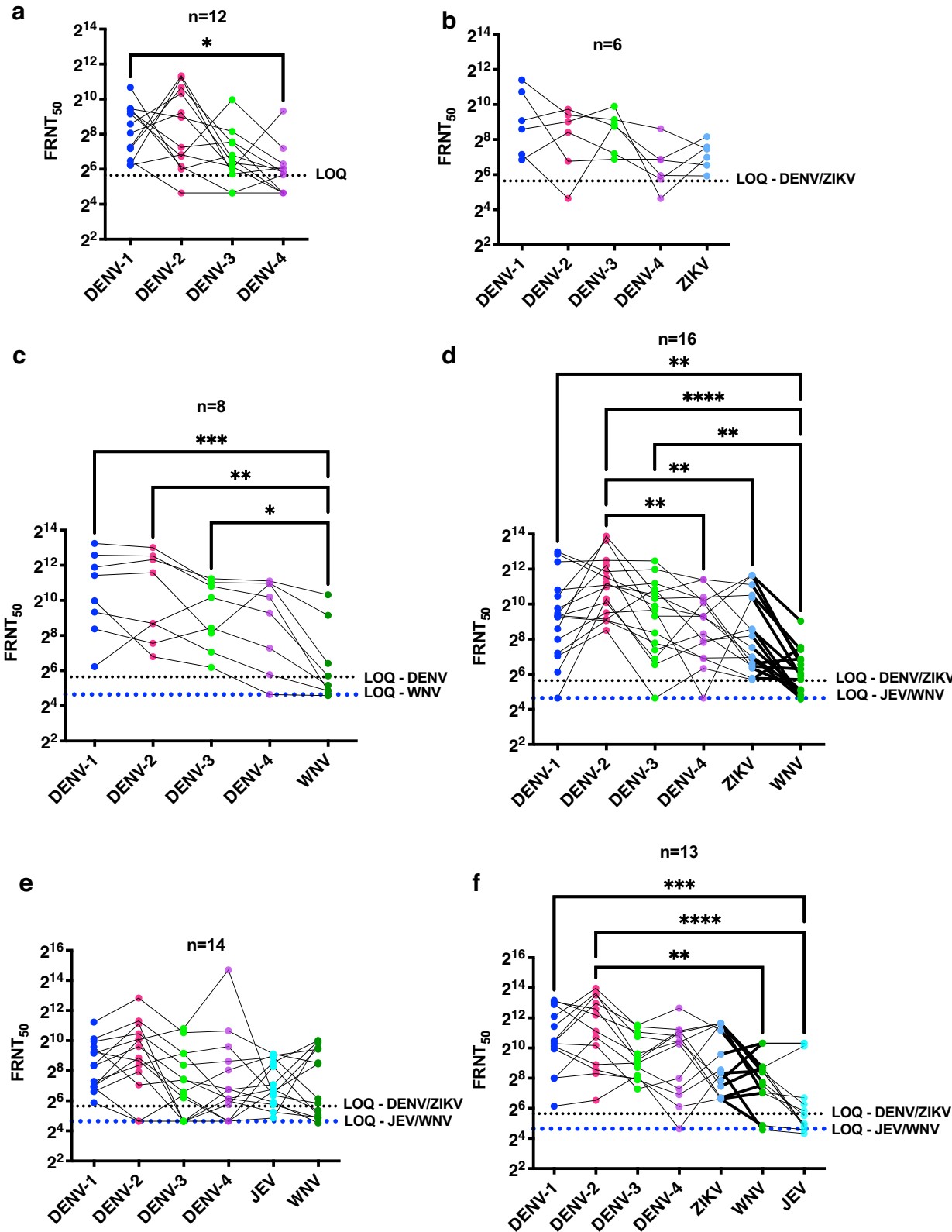

for the past few years[20,43] as compared to the New Guinea C strain of DENV-2 which belongs to Asian genotype. DENV-3 isolate from India belongs to genotype III while the international reference strain from Indonesia is from genotype I[43]. Similar to our previous report, the DENV-4 isolate from India belongs to genotype I while the international DENV-4 isolate is from genotype II[43]. Therefore, we were interested in assessing whether the neutralizing antibodies elicited against the Indian circulating viruses can neutralize the international DENV isolates. We selected 33 random samples which had detectable neutralizing antibodies to all the four DENV serotypes and tested the ability of these samples to neutralize international DENV isolates. We observed that neutralizing ability of antibodies was reduced for international DENV-1, 3 and 4 serotypes as compared to Indian

**Fig. 5 Flavivirus cross-reactive neutralizing antibody responses.** Cross reactivity of dengue neutralizing antibodies to related flaviviruses, namely, DENV-1 (blue circles), DENV-2 (red circles), DENV-3 (light green circles), DENV-4 (purple circles), ZIKV (sky blue circles), JEV (cyan circles) and WNV (dark green circles) was estimated by FRNT assay in a subset of samples. **a** Subset of independent samples positive for neutralizing antibodies to one or more DENV serotypes ($n = 12$). In all the panels, mean rank of each column was compared with mean rank of every other column using paired, non-parametric, Friedman test. Correction for multiple comparisons was applied by Dunn's multiple comparison test. *$P = 0.0123$. **b** Subset of independent samples positive for neutralizing antibodies to DENV and ZIKV ($n = 6$). **c** Subset of independent samples positive for neutralizing antibodies to DENV and WNV ($n = 8$). ***$P = 0.0004$; **$P = 0.0015$; *$P = 0.0266$. **d** Subset of independent samples ($n = 16$) positive for neutralizing antibodies to DENV, ZIKV and WNV ($n = 16$). ****$P < 0.0001$; **$P = 0.0016$ (DENV-1 vs WNV); **$P = 0.0071$ (DENV-2 vs DENV-4); **$P = 0.0049$ (DENV-2 vs ZIKV); **$P = 0.0024$ (DENV-3 vs WNV). **e** Subset of independent samples positive for neutralizing antibodies to DENV, WNV and JEV ($n = 14$). **f** Subset of independent samples positive for all the flaviviruses indicated in the figure ($n = 13$). ****$P < 0.0001$; ***$P = 0.0006$; **$P = 0.0050$. LOQ (Limit of quantitation).

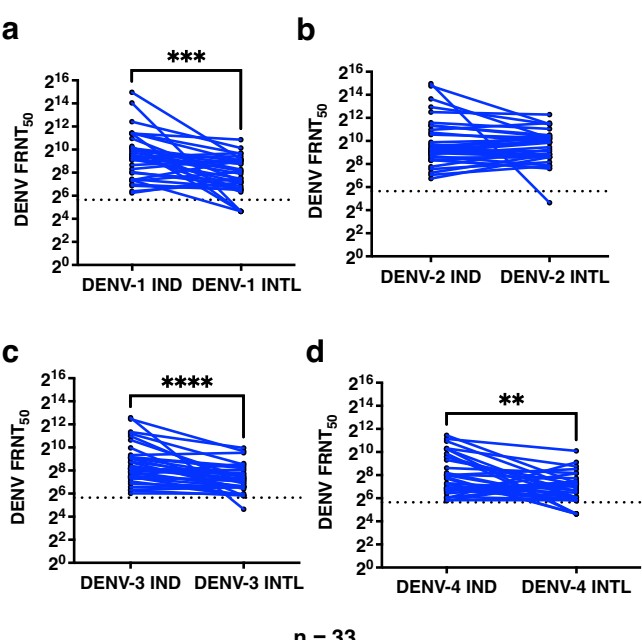

n = 33

**Fig. 6 Cross reactivity of dengue neutralizing antibodies to global strains of DENV-1-4.** (**a**–**d** respectively) was estimated by FRNT assay in a subset of samples ($n = 33$ biologically independent samples). DENV-IND indicates Indian isolates of DENV1-4 whose accession numbers are provided in the methods section. DENV-INTL indicates the international isolates of DENV namely rDENV1 WP (DEN1-Western Pacific), DENV-2 (New Guinea-C prototype strain), rDENV3-7164 (DEN3-Sleman/78) and rDEN4 7-4A-1A2 (DENV4-Dominica/81). Significance between the observations were determined by Wilcoxon matched-pairs signed rank test. Two-tailed *P*-value is shown. ****$P < 0.0001$; ***$P = 0.0001$; **$P = 0.0033$. LOQ (Limit of quantitation).

**Table 4 FRNT$_{50}$ titers for Indian DENV isolates and International DENV strains measured in a subset of samples ($n = 33$).**

| Virus used for neutralization | Geometric mean (95% CI) | | P-value* |
| --- | --- | --- | --- |
| | IND (Indian isolate) | INTL (International isolate) | |
| DENV-1 | 711 (428–1183) | 217 (144–325) | 0.0001 |
| DENV-2 | 926 (554–1549) | 708 (492–1020) | 0.3960 |
| DENV-3 | 386 (253–588) | 149 (111–200) | <0.0001 |
| DENV-4 | 256 (168–390) | 109 (79–151) | 0.0033 |

*Wilcoxon matched-pairs signed rank test.
IND indicates Indian isolates of DENV1-4 whose accession numbers are provided in the methods section. INTL indicates the international isolates of DENV namely rDENV1 WP (DEN1-Western Pacific), DENV-2 (New Guinea-C prototype strain), rDENV3-7164 (DEN3-Sleman/78) and rDEN4 7-4A-1A2 (DENV4-Dominica/81).

isolates (Fig. 6a, c and d and Table 4). Consistent with higher levels of neutralizing antibodies to DENV-2, there was no significant difference between the neutralization antibody titers for DENV-2 Indian and International isolates (Fig. 6b and Table 4). Therefore, it is desirable to compare the neutralizing antibodies for their ability to neutralize both the international and circulating dengue isolates in vaccine efficacy studies that are being planned in India and elsewhere.

## Discussion

The post-market safety data of Dengvaxia has led to WHO recommending vaccination of this vaccine only in seropositive subjects. A second live-attenuated dengue vaccine, TAK-003, has been approved in Indonesia in subjects regardless of their seropositive status (https://www.takeda.com/newsroom/

newsreleases/2022/takedas-qdenga-dengue-tetravalent-vaccine-live-attenuated-approved-in-indonesia-for-use-regardless-of-prior-dengue-exposure/). Another live-attenuated vaccine developed by the National Institutes of Health is currently undergoing advanced stage clinical trials in developing countries and Phase III trials are anticipated to begin soon in India [44,45]. Previous studies have shown that the neutralizing antibody is a good correlate of protection from dengue disease[46, 47]. These studies from Nicaragua and Thailand have estimated a cut-off for neutralizing antibody titers that associate with protection from re-infection. However, no such data exists for Indian population and considering the current plans to initiate vaccine efficacy studies in India, it is imperative that more data is needed to inform vaccine trials. As a large proportion of Indian adults are seropositive, it is essential to estimate the prevalence of dengue and repertoire of neutralizing antibodies at various geographic locations within India. We show that over 72% of the acute febrile illness patients were positive for dengue IgG by indirect ELISA which indicates a prior exposure in these cases. About 69% of the samples tested for neutralizing antibodies had a multitypic response and the titers for DENV-2 was highest which was similar to the observations from a cohort in the city of Pune in India[18]. In agreement with this study, we also observed that the titers of DENV-4 antibodies were the lowest compared to the other three DENV serotypes. The question of whether DENV-4 infections are less prevalent or whether DENV-4 is more susceptible to neutralization and restriction by pre-existing immunity needs further investigation. The profile of neutralizing antibodies vary based on the geographical region within India and provide clues to circulating serotypes in the region[17]. Higher levels of neutralizing antibodies were observed for DENV-1, DENV-2 to DENV-3 serotypes which were the predominant circulating serotypes in the past few years[30, 32, 43]. Although replication fitness in mosquitoes plays a determining

role in selection of circulating viruses[48], we speculate that the pre-existing neutralizing antibodies may also exert selection pressure and would give rise to DENV isolates that predominate every season. Previous report from Bangkok suggests immunological cross-reaction between the serotypes as a prominent driver of clade extinction and serotype prevalence[49]. A combination of prior immunity and clade replacement has been shown to contribute to severe dengue in a cohort study from Nicaragua[50]. In addition, immunological interaction between DENV-1 and DENV-4 was predicted in unusual phase separation of DENV-4 outbreaks[49]. Similar clade replacements have been reported from dengue endemic countries and cross-protective immune responses in the population have been shown to drive virus evolution and serotype dominance[50, 51]. We are observing a similar pattern here in India where DENV-4 appears to be the least circulating serotype in the study region and the lowest levels of neutralizing antibodies are against DENV-4 indicating a potential for next DENV-4 outbreak and dominance. As detection of circulating serotypes is biased towards serotypes that are more likely to result in symptomatic disease and people seeking care, it is also possible that DENV-4 infections are mild or asymptomatic in the study area with high levels of pre-existing immunity and, therefore, there is minimal detection of DENV-4 cases and low levels of DENV-4-specific antibodies elicited in the study participants. However, dengue outbreaks with DENV-4 as the predominant serotype has been reported from Southern part of India[52]. India reported its first indigenous Zika case in November 2016 and, ever since, Zika outbreaks are reported from different parts of India and a distinct Asian lineage of ZIKV has been identified as the circulating strain[53–55]. We observed about half of the 76 DENV FRNT positive samples were also neutralizing ZIKV suggesting either silent circulation of ZIKV or repeated exposure to DENV eliciting high levels of cross-neutralizing antibodies. Other flaviviruses such as JEV and WNV circulate in certain pockets of India, however, we observed significantly low levels of cross-neutralizing antibodies to these two related flaviviruses suggesting lack of exposure to these viruses in the study region.

We and others have recently characterized the evolutionary trajectory of circulating DENV isolates from India and showed that the Indian isolates are uniquely clustered as a separate genotype[40–43]. The envelope region of the circulating Indian dengue virus isolates diverge significantly from the vaccine strains, either licensed or under late stage clinical trials. However, our data shows that the antibodies elicited in response to natural dengue infection in India were capable of neutralizing international dengue isolates albeit at a lower level for DENV-1, DENV-3 and DENV-4. Nevertheless, based on efficient neutralization of an international DENV-2 isolate we speculate that higher level of antibodies may show similar effect for the other three international serotypes. Our observations with DENV-2 neutralization is similar to that of Thai cluster studies where the strain of DENV-2 used in plaque reduction neutralization titer assays made no difference to the estimation of association of neutralizing antibodies with protection from DENV infection[46]. Although the vaccine candidates that are currently under development in India are generated from viruses that were in circulation almost two decades ago, we predict that vaccination with any of the dengue vaccine candidates under development may further boost the existing level of antibodies in seropositive individuals which would be able to neutralize both Indian and International dengue strains. Antibody levels have been proposed as a reliable correlate of protection for dengue based on longitudinal cohort studies[46, 47]. However, the phenomenon of ADE has obligated clinical outcomes as a necessary end-point for vaccine efficacy[56]. In the context of COVID-19, we had

shown that high levels of pre-existing antibodies showed a negative correlation with boosting effect on antibody titers upon vaccination with an inactivated COVID-19 vaccine[57]. For Phase III trials of dengue vaccine, it may be necessary to assess whether individuals with high levels of pre-existing antibodies to dengue would be suitable for demonstrating the rise in neutralizing antibody titers upon vaccination and protection as these subjects, if enrolled into the study, may be protected from infection even otherwise due to high levels of pre-existing antibody titers as demonstrated by previous studies[47].

Some of the limitations of our study are that the samples received were anonymized and without information on the clinical outcomes. Therefore, we were not able to deduce any association of infecting serotype, viremia or neutralizing antibody titers with clinical outcomes. The samples were collected at a single time point during the acute phase of the illness and although over 90% of the samples were positive for viral RNA indicating early stages of illness, the day of fever at the time of diagnosis could be a variable contributing to some of the observed differences in viremia. Another limitation was the non-availability of sufficient volume of seropositive samples which did not allow us to test all the samples in neutralization assays. Our ongoing efforts are focussed on monitoring the circulating virus strains and to quantify changes in the profile of neutralizing antibody titers in the population by longitudinal cohort studies which will further inform policy on vaccine roll out for dengue in India.

## Data availability
All the data are presented in this manuscript and in the supplementary information. Supplementary Data 2 contains source data for the main figures with numerical values in this paper. The sequencing files for the Indian DENV isolates used in this study were accessed with the GenBank reference numbers: ON799266, ON799267, ON799401 and OP310810.

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

## Acknowledgements

We thank all the members of the bioassay lab for technical support. We thank Dr. Deepti Nariani from SRL diagnostics for coordinating sample transfers. This work was supported by the Department of Biotechnology (DBT) through IndCEPI Mission (BT/MB/CEPI/2016), Translational Research Program (BT/PR30159/MED/15/188/2018). We also acknowledge funding support from the Biotechnology Industry Research Assistance Council (BIRAC)-National Biopharma Mission (BT/NBM0099/02/18). The funders had no role in study design, data collection and interpretation or the decision to submit the work for publication.

## Author contributions

A.A., T.A., A.T., P.K., I.K., and M.P. performed all the serological, molecular and neutralization assays. P.S., B.S., C.P., S.K. and R.P. were involved in virus sequencing efforts and analysis of whole genome sequencing data. A.C., R.L. and S.S.W. contributed resources, provided critical inputs in experimental design and data analysis. GRM conceived the study, designed the experiments and analyzed the data. A.A., T.A., and G.R.M. wrote the first draft of the manuscript. G.R.M. wrote the final version. All authors have reviewed and approved the final submitted version of the manuscript.

## Competing interests

The authors declare no competing interests.

## Additional information

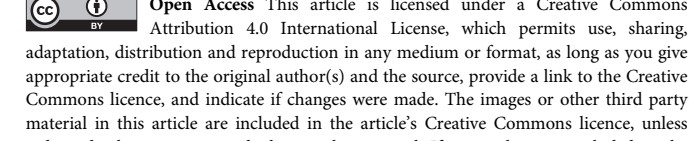

