## [Peer Review File · Communications Medicine]

Reviewers' comments:

Reviewer #1 (Remarks to the Author):

Synopsis: This study is an analysis of samples (n=412) from suspected DENV cases collected in 2018-2019 from the region around Delhi, India (a region of approximately 50,000 km²). Viral RNA levels were quantified for the majority of samples, as was anti-DENV IgM, and IgG. Viral RNAemia negatively correlated with IgG titers in secondary infections. Primary infections (IgM:IgG>1.8) generally showed higher levels of viremia. The authors then determine infecting serotypes from most samples as well as neutralizing titers of these samples against all 4 serotypes. They further note that neutralizing titers differ significantly within serotypes when sera from their cohort were tested against Indian and "international" isolates of the same serotype.

In general, this is a comprehensive analysis of the serology associated with a DENV-endemic area during a period in which all 4 serotypes were circulating. The manuscript is clearly written and easy to follow. As the authors note, the study would be significantly strengthened by information on the severity of disease associated with each patient. Unfortunately, this information was not available. One minor point – on line 216, page 9, the authors describe ZIKV as belonging to the same serogroup as DENV. Although anti DENV and anti-ZIKV sera do cross-react, ZIKV is classified as part of the Spondweni antigenic serogroup, not DENV.

Reviewer #2 (Remarks to the Author):

The authors reported on lower viremia levels in patients with higher neutralizing antibody levels to dengue. While the data is of importance, there are a number of publications on higher neutralizing levels that is correlated with lower viremia levels. The authors would need to add more data to demonstrate novelty including that of outcomes, ie clarify/quantify the protective levels of antibodies and include ADE activity in the determination of neutralizing levels against DENV.

Reviewer #3 (Remarks to the Author):

In this study, A. Anantharaj, et al. evaluated the antibody levels and RNA copies in Dengue patient sera in India. In general, the authors conducted a lot of experiment. The manuscript is well-presented and contains important issues. The data are often well described, but the conclusions and considerations that can be made from the results are a bit ambiguous (especially Figure 4).

My specific comments are described below.

Figure 2 D and E

The authors might want to show the data of DENV RNA copy numbers in primary infections and secondary infections categorized by age groups.

Line 183, The authors stated that "The lower levels of FRNT50 titers observed for DENV-3 and DENV-4 coincided with DENV-3 and DENV-4 being the predominant infecting serotype in these subjects (Fig. 4C)." However, the reviewer could not understand the relation between the lower levels of FRNT50 titers and being the predominant infecting serotype. It would be better to address this in

more detail.

Line 206, The authors stated that “Thus the lower levels of neutralizing antibodies in primary infections correlate with higher viremia.” Please state that the which data made the authors conclude this.

Line 245, The authors stated that “Consistent with higher levels of antibodies to DENV-2, there was no significant difference between the neutralization antibody titers for DENV-2 Indian and International isolates (Fig. 5C and Table 5).” However, the reviewer could not understand the relation between the higher levels of antibodies and no significant difference between the neutralization antibody titers. Again, it would be better to address this in more detail.

Line 278, “Similar reports of clade replacements have shown been shown” Is this typo?

Response to reviewers

Reviewers' comments:

Reviewer #1 (Remarks to the Author):

Synopsis: This study is an analysis of samples (n=412) from suspected DENV cases collected in 2018-2019 from the region around Delhi, India (a region of approximately 50,000 km²). Viral RNA levels were quantified for the majority of samples, as was anti-DENV IgM, and IgG. Viral RNAemia negatively correlated with IgG titers in secondary infections. Primary infections (IgM:IgG>1.8) generally showed higher levels of viremia. The authors then determine infecting serotypes from most samples as well as neutralizing titers of these samples against all 4 serotypes. They further note that neutralizing titers differ significantly within serotypes when sera from their cohort were tested against Indian and “international” isolates of the same serotype.

In general, this is a comprehensive analysis of the serology associated with a DENV-endemic area during a period in which all 4 serotypes were circulating. The manuscript is clearly written and easy to follow. As the authors note, the study would be significantly strengthened by information on the severity of disease associated with each patient. Unfortunately, this information was not available. One minor point – on line 216, page 9, the authors describe ZIKV as belonging to the same serogroup as DENV. Although anti DENV and anti-ZIKV sera do cross-react, ZIKV is classified as part of the Spondweni antigenic serogroup, not DENV.

Response: We thank the reviewer for the positive comments. Thank you for pointing out the error in serogroup of ZIKV. We have corrected this in the revised manuscript.

Reviewer #2 (Remarks to the Author):

The authors reported on lower viremia levels in patients with higher neutralizing antibody levels to dengue. While the data is of importance, there are a number of publications on higher neutralizing levels that is correlated with lower viremia levels. The authors would need to add more data to demonstrate novelty including that of outcomes, ie clarify/quantify the protective levels of antibodies and include ADE activity in the determination of neutralizing levels against DENV.

Response: We thank the reviewer for the comments. As stated in our manuscript - India is endemic to dengue and contributes to about one-third of the global dengue cases, and we show that over 70% of adults have prior exposure to Dengue. However, there are very few studies that have addressed the association between pre-existing antibodies and viremia during acute phase of infection from Indian population. There is no data on association between infecting serotype and prior immunity to that serotype. In addition, in this manuscript, we also demonstrate varying degree of neutralization capacity of antibodies generated from natural exposure to dengue to Indian and International DENV isolates. All these observations are novel and provide a foundation to justify further investment on data generation to assist/facilitate policy on vaccination and dengue management.

While we appreciate that analysis of ADE responses would be of academic interest, we would like to emphasize that the neutralizing antibody is considered as a gold-standard correlate of protection from disease/infection (Proc Natl Acad Sci U S A 2016; 113(3): 728-33; DOI: [10.1371/journal.pntd.0003230](https://doi.org/10.1371/journal.pntd.0003230)). These studies from Nicaragua and Thailand have estimated cut-off of neutralizing antibody titers that associate with protection from re-infection. However, no such data exists for Indian population and considering the current plans to initiate vaccine efficacy studies in India, it is imperative that more data is needed to inform vaccine trials. We show that a large proportion (over 70%) of adults in India have neutralizing antibodies to all the four serotypes of dengue virus which suggests that over half of the adult population is unlikely to experience severe dengue due to the presence of high levels of neutralizing antibodies. While many studies have further characterized the epitopes that can confer serotype specific or cross-protective responses by isolating monoclonal antibodies, to decipher the mechanism of neutralization versus ADE functions ([doi:10.1128/JVI.00273-15](https://doi.org/10.1128/JVI.00273-15)), it is beyond the scope of our study as we only obtained minimal amount of serum samples from patients with acute febrile illness from the diagnostic lab. Moreover, because the sera will contain a mix of antibodies with variable neutralization/ADE functionality, it would not provide any definitive answer to the relative association of ADE and neutralization activity to the observed viremia.

Only about 10-15% of the secondary dengue patients develop severe disease. Studies from long-term paediatric cohort of dengue in Nicaragua has shown that children with pre-existing antibody titers of 1:20 to 1:80 are at a higher risk of developing severe dengue as compared to children with antibody titers lower than 1:20 suggesting that waning antibody titers play a major role in predisposing infected individuals towards developing severe dengue in secondary infections due to ADE. Also, studies in infants where antibodies collected at 6 months of age showed peak in infection enhancement in in vitro ADE assays ([J Infect Dis. 2008 Aug 15; 198\(4\): 516–524](https://doi.org/10.1128/JID.198.4.516-524)). Our study does not permit us to do such analysis because the samples are anonymized which we have stated as a limitation in our manuscript. For a meaningful interpretation of data from in vitro ADE assays, we would have to establish a longitudinal, community-based, cohort studies similar to the ones in Thailand, Vietnam or Nicaragua which is currently not feasible and will also require considerable amount of time and resources.

Nevertheless, to address the queries of Reviewer 2 and the editor, we tested for ADE of DENV infection in K562 cells which is the model cell line used by most of the studies to demonstrate ADE because it is not permissive to dengue infection in the absence of Fc mediated uptake of virus. Plasma samples collected from six paediatric dengue patients with secondary infections (less than five days of fever) were used to test for enhancement of infection in vitro in K562 cells. We are presenting this data as a response to both editor and reviewer's concerns in the figure below. As shown in the figure below, every sample demonstrates enhancement of infection of all the four serotypes albeit at different dilutions which suggest that the plasma/serum samples with a complex mix of polyclonal antibodies may not yield interpretable data in this assay to determine the role of ADE in the observed viremia.

Figure: *Antibody-dependent enhancement of DENV infection.* Plasma samples from dengue patients collected at the time of enrolment (<7 days of fever) were serially diluted as indicated in the x- axes and incubated for one hour at 37°C with DENV1-4 respectively and added on to K562 cells. The unbound immune complexes were removed and cells were washed and cultured in normal growth medium for 24 hours and stained for DENV envelope by using pan-DENV primary antibody (4G2) conjugated with Alexa Fluor dye. % of infection was determined by flow cytometry.

Reviewer #3 (Remarks to the Author):

In this study, A. Anantharaj, et al. evaluated the antibody levels and RNA copies in Dengue patient sera in India. In general, the authors conducted a lot of experiment. The manuscript is well-presented and contains important issues. The data are often well described, but the conclusions and considerations that can be made from the results are a bit ambiguous (especially Figure 4).

My specific comments are described below.

Figure 2 D and E

The authors might want to show the data of DENV RNA copy numbers in primary infections and secondary infections categorized by age groups.

Response: We have revised figure 2E and shown data categorized by age groups in primary and secondary infections in new Figure 2E and 2F respectively.

Line 183, The authors stated that “The lower levels of FRNT50 titers observed for DENV-3 and DENV-4 coincided with DENV-3 and DENV-4 being the predominant infecting serotype in these subjects (Fig. 4C).” However, the reviewer could not understand the relation between the lower levels of FRNT50 titers and being the predominant infecting serotype. It would be better to address this in more detail.

Response: Our hypothesis was that the level of neutralizing antibodies, either homotypic or heterotypic, may determine the susceptibility to infection with circulating DENV serotype. Individuals with antibodies to any of the serotypes may be protected from reinfection either due to homotypic protection or cross-protection unless the antibody levels are below the level required for protection or the infecting serotype escapes pre-existing humoral immunity. More than 60% of the samples were positive for DENV-3 and the lowest levels of neutralizing antibodies observed was against DENV-3 and DENV-4. We have clarified this in the discussion section in lines 178-182 and lines 279-284.

Line 206, The authors stated that “Thus the lower levels of neutralizing antibodies in primary infections correlate with higher viremia.” Please state that the which data made the authors conclude this.

Response: We observed a significantly higher viremia in primary dengue infections (Fig. 2D) compared to secondary dengue. We next segregated FRNT data from 215 samples as per the primary (n=45) or secondary infection (n=170) status. Thirty two of the 45 primary dengue samples showed no neutralizing antibodies to any of the four serotypes. Of the remaining thirteen samples, ten samples had antibodies to any one of the DENV serotype and three samples had multitypic neutralizing antibody response (Fig. 4I). This is described in lines 213-218.

Line 245, The authors stated that “Consistent with higher levels of antibodies to DENV-2, there was no significant difference between the neutralization antibody titers for DENV-2 Indian and International isolates (Fig. 5C and Table 5).” However, the reviewer could not understand the relation between the higher levels of antibodies and no significant difference between the neutralization antibody titers. Again, it would be better to address this in more detail.

Response: We regret the typo. The titers of “neutralizing” antibodies was highest for DENV-2 in the samples. These levels of neutralizing antibodies may be high enough to neutralize both the Indian and International isolates despite the viruses being divergent from each other.

Line 278, “Similar reports of clade replacements have shown been shown” Is this typo?

Response: We regret the typo. This has been rectified.

REVIEWERS' COMMENTS:

Reviewer #1 (Remarks to the Author):

The revised version of this manuscript addresses my previous concerns.

Response to reviewers

Reviewers' comments:

Reviewer #1 (Remarks to the Author):

The revised version of this manuscript addresses my previous concerns.

Response: We thank the reviewer for sparing his/her valuable time in providing us constructive comments which helped us to improve our manuscript.